# Research on Metal Corrosion Resistant Bioinspired Special Wetting Surface Based on Laser Texturing Technology: A Review

**DOI:** 10.3390/mi13091431

**Published:** 2022-08-30

**Authors:** Li Zhang, Zheng Tan, Chong Zhang, Jingrong Tang, Chi Yao, Xiangyu You, Bo Hao

**Affiliations:** 1School of Control Engineering, Northeastern University at Qinhuangdao, Qinhuangdao 066004, China; 2Key Laboratory of Vibration and Control of Aero-Propulsion System Ministry of Education, Northeastern University, Shenyang 110819, China

**Keywords:** metal substrates, laser texturing, corrosion resistance, bioinspired special wetting surface

## Abstract

Metal substrates are widely used in engineering production. However, material life reduction and economic loss due to chemical and electrochemical corrosion are a major problem facing people. Electrochemical corrosion is the main corrosion mode of metals, such as seawater corrosion. It is found that the superhydrophobic surface treated by laser texturing plays an important role in the corrosion resistance of the substrate, with the laser texturing process and post-treatment affecting the corrosion resistance. The corrosion resistance is positively correlated with the superhydrophobic property of the surface. For the mechanism of corrosion resistance, this paper summarizes the effect of micro-nano structure, surface-modified coating, oxidation layer or new product layer, surface inhomogeneity, crystal structure, and slippery surface on corrosion resistance. Superhydrophobic surface and slippery surface are two common types of bioinspired, special wetting surfaces. In order to prepare better superhydrophobic and corrosion-resistant surfaces, this paper summarizes the selection and optimization of laser parameters, surface structure, processing media, and post-treatment from the point of view of mechanism and law. In addition, after summarizing the corrosion resistance mechanism, this paper introduces a series of characterization experiments that can measure the corrosion resistance, providing a reference for preparation and evaluation of the surface.

## 1. Introduction

Metal materials are widely used as substrates in daily life. However, metal corrosion has always been a problem that puzzles people. Superhydrophobic surfaces prepared by laser texturing are resistant to corrosion due to their unique surface structure and oxide layer. On this basis, surface post-treatments such as surface modification and lubricant filling can further enhance the ability and are of great significance in the research of corrosion resistance. In recent years, research on the fabrication of bioinspired special wetting surfaces on metal substrates and the further development of functional surfaces with corrosion resistance has become an important method to solve this problem. Superhydrophobic surface and slippery surface are two common types of bioinspired special wetting surfaces. To fabricate superhydrophobic surfaces, there are two key points. On the one hand, micro-nano rough structures are formed to accommodate an air cushion. On the other hand, reducing the free energy of the surface makes it difficult for the droplets to expand on the surface. Common processing methods for preparing superhydrophobic surface are laser texturing [1], electrochemical deposition [2], chemical etching [3], hot embossing [4], chemical vapor deposition [5], sol–gel method [6], hydrothermal method [7], and spraying method [8]. Common corrosion resistance processing methods are shown in Table 1. However, most of the methods require large-scale instruments and expensive preparation equipment, while the steps are complex, the chemical preparation process is long, the process procedures are many, the generation period is long, and the stability and controllability are poor [9]. By contrast, due to the advantages of good economy, high processing efficiency, convenient processing, large processing scale, low pollution, and strong controllability, laser texture processing is increasingly widely used.

Micro-nano rough structures are usually constructed by abstracting and imitating bionic structures. There are many examples of superhydrophobic and slippery surfaces in nature, such as lotus leaves [10], water repellent legs [11], fish scales [12], reed leaves [13], and cabbage [14]. The geometric structure is abstracted from the microelectronic structure of these natural superhydrophobic surfaces, with the structure manufactured by laser texture processing. Univariate experiments are carried out on the laser physical parameters and processing parameters in the processing to determine the most appropriate parameter allocation scheme for different materials. After texture processing, surface modification and lubricant injection are considered to prepare functional surfaces with corrosion resistance. The factors affecting the corrosion resistance and stability are: surface air cushion structure, resistance of modifier coating to charge transfer [14], synergistic effect of oxide layer, surface nonuniformity, crystal structure and composition, good self-healing and droplet retention resistance of slippery surfaces, and physical damage resistance of micro-nano composite structure [15], while the degree and reason of improving the corrosion resistance are evaluated through relevant characterization experiments. At present, the corrosion resistance performance based on a superhydrophobic surface has been applied in many fields and has strong development potential.

Firstly, this paper introduces the relevant theories of superhydrophobic surface and slippery surfaces preparation, including the theoretical model, formation mechanism, and the preparation and application of the slippery surface related to its corrosion resistance function. Secondly, this paper analyzes the related theories of laser texture processing, including laser type and parameters, processing medium, texture structure, etc. Finally, the corrosion resistance mechanism, corrosion resistance treatment process, and corrosion resistance characterization experiment are summarized, with the existing problems and development direction outlined. Table 2 outlines the meaning of all variables referred to in this article.

## 2. Superhydrophobicity Mechanism

### 2.1. Contact Angle (CA) and Rolling Angle (RA) Theory

The contact angle (CA) is one of the important parameters used to evaluate surface affinity. When droplets reach the solid surface, when the solid–liquid–gas phase reaches a stable state, the droplets will show a crowned geometry, making a tangent from the solid–liquid–gas three-phase contact. As shown in Figure 1, the angle between the gas phase line and the solid–liquid line is the size of the contact angle, which is recorded as θ. If θ≈0°, the surface is called a superhydrophilic surface; if θ<90°, the surface is called a hydrophilic surface; if 90°<θ<150°, the surface is called a hydrophobic surface; if θ>150°, the surface is called a superhydrophobic surface, as shown in Figure 2.

In addition, the hydrophobic degree of the surface can be expressed by combining the advance contact angle and the retraction contact angle. When the contact angle changes because of an increase in droplets, the contact angle formed after the liquid–solid interface replaces the gas–solid interface is called the advance angle, and is recorded as $\theta_{A}$, as shown in Figure 3a. Similarly, the contact angle formed after the gas–solid interface replaces the liquid–solid interface is called the retraction angle, and is recorded as $\theta_{R}$, as shown in Figure 3b. $\theta_{A}-\theta_{R}$ is the contact hysteresis angle. The sliding ability of the surface droplets can be judged by the contact hysteresis angle.

The rolling angle (RA) is another important parameter for evaluating surface wettability. The rolling angle is the inclination angle at which the surface of the object is tilted at a slow angular speed to allow the droplet to begin rolling, as shown in Figure 3c. The rolling angle indicates the viscous force of the droplet on the surface. The smaller the rolling angle, the weaker the viscous force. In many viewpoints, superhydrophobic surfaces require contact angles of more than 150° and rolling angles of less than 10°.

### 2.2. Theoretical Model of Hydrophobicity

Young [17] proposed the relationship between the surface tension and contact angle of water droplets on a slippery and uniform ideal surface, as shown in Figure 4a. The formula showing the relationship between surface tension and contact angle is:(1)$$\cos\;\theta =\frac{\gamma_{sg}-\gamma_{sl}}{\gamma_{gl}}$$
where $\gamma_{sg}$ is the surface tension of the solid–gas interface, $\gamma_{sl}$ is the surface tension of the solid–liquid interface, and $\gamma_{gl}$ is the surface tension of the gas–liquid interface. The formula describes the functional relationship between the solid–gas surface tension, solid–liquid surface tension, gas–liquid surface tension, and contact angle. However, because Young’s model is a function model of an ideal surface and will not appear in real production, it is closer to the theoretical model and is rarely used in practice [18,19,20].

The Wenzel [21] model is based on a rough surface, considering the influence of the actual rough fabricating on the contact angle. The rough surface is modeled as several small grooves, and it is assumed that the droplets can completely immerse into the groove, as shown in Figure 4b, which has a certain impact on the contact angle [22]. The Wenzel formula is.
(2)$$\cos\theta_{r}=r(\frac{\gamma_{sg}-\gamma_{sl}}{\gamma_{gl}})=r\cos\theta_{e}$$
where r is the surface roughness factor, which is equal to the ratio of the actual contact area between solid and liquid to its apparent contact area, and is always greater than 1; $\theta_{r}$ is the contact angle of the droplet on the solid rough surface, which is called the apparent contact angle; and $\theta_{e}$ is the contact angle of the droplet on the ideal surface of the solid, which is called the intrinsic contact angle. For hydrophilic surfaces, the surface roughness increases and the hydrophilicity increases. When the intrinsic contact angle of the material is greater than 90°, the surface is originally hydrophobic. Increasing the surface roughness can enhance its hydrophobicity. However, this model is usually only applicable to the uniform solid–liquid interface and has certain limitations on the solid–liquid–gas interface.

The Cassie–Baxter [23] model is a three-phase contact of solid, liquid, and gas, as shown in Figure 4c. The air in the surface groove prevents the liquid from penetrating into the groove, thereby forming a composite contact surface. The Cassie formula is:(3)$$\cos\theta_{r}=rf_{1}\cos\theta_{e}-f_{2}$$
where $f_{1}$ is the proportion of solid–liquid contact area in composite contact area, $f_{2}$ is the proportion of gas–liquid contact area in the composite contact area, and $f_{1}+f_{2}=1$. According to the contact formula, the contact angle is not only related to the intrinsic contact angle and surface roughness, but also related to the proportion of solid–liquid and gas–liquid contact. The larger the proportion of gas and liquid, the larger the contact angle obtained. Therefore, the more gas the air cushion contains, the more hydrophobic the surface is. The Cassie–Baxter model is now recognized as the most scientific model. This model is the basis for the study and exploration of superhydrophobic surfaces [24]. Partaker et al. [25] explored the selection principle between the Wenzel model and the Cassie model through the energy local minimum theory of equilibrium droplets on rough substrates. In the selection of theoretical models, Bhushan et al. [26] found that when the distance between the lowest liquid level column and the top of the cylinder is far greater than the height H of the cylinder array, the liquid contacts the bottom of the trench, the surface cannot store air, and the surface shape will be converted to the Wenzel model. The calculation formula is:(4)$$\frac{{\left(\sqrt{2}P-D\right)}^{2}}{R}\geq H$$
where P is the pitch, D is the diameter, R is the radius of curvature, and H is the height, as shown in Figure 5.

In addition to the three classical models described above, Herminghaus also provides a new model. The Herminghaus model [27] is a model concept proposed after studying the leaves of hydrophobic plants in nature on the basis of the Cassie model. In order to obtain more air cushions, more gas can be accommodated through the slot structure with hierarchical structure to form a larger contact angle, as shown in Figure 6.

Under this hierarchical structure, the Herminghaus formula is
(5)$$\cos\theta_{n+1}=(1-\omega_{n})\cos\theta_{n}-\omega_{n}$$
where n is the series, $\theta_{n}$ is the apparent contact angle of the surface of the nth stage structure, and $\omega_{n}$ is the proportion of gas–liquid contact area on the surface of the nth stage structure. It can be seen from the function that the more orders are generated, the greater the contact angle generated. Thus, if the intrinsic contact angle is greater than zero, the more complex the hierarchical structure that is made on the substrate surface. If the droplet can be suspended on the smallest structure, the surface is superhydrophobic. Few researchers have used the Herminghaus model as their theoretical basis in previous studies. This model with excellent superhydrophobic ability should be paid more attention.

### 2.3. Formation of Superhydrophobic Surface

Constructing a superhydrophobic surface is the basis of preparing an anticorrosive functional surface. Researchers can better discover the mechanism of corrosion resistance from the causes of superhydrophobic properties, so it is necessary to discuss the law of wettability change. However, the reasons for the change of surface wettability are still under discussion. In addition to the Wenzel model [28], the reasons why some surfaces become super hydrophilic after laser ablation also involve the generation of super hydrophilic oxides in the outer layer during the manufacturing process [29]. Yang et al. [8] supports this theory. They also believed that this was related to the high degree of polarity of the surface. In the process of laser irradiation, due to the carbonization reaction, the alkylates contained in the matrix itself are greatly damaged, the polarity of the surface increases, and the hydrophilicity increases [30].

The reasons why other surfaces become hydrophobic after laser treatment are not uniform. Some researchers believe that the treated substrate will produce new superhydrophobic substances to protect the substrate. For example, active magnetite is produced under the energy brought by laser radiation. This magnetite can decompose carbon dioxide in the air into carbon, which is hydrophobic, to achieve the hydrophobic effect [31]. Laser irradiation of a copper surface will produce a deoxidation reaction on the surface, and the cuprous oxide generated is hydrophobic [32]. In addition, the surface of titanium oxide can protect the titanium alloy matrix [33]. Iakovlev et al. [34] formed a superhydrophobic oxide film on the surface of steel and titanium, improving the surface hydrophobicity. Another theory is that the lower the surface free energy, the higher the degree of nonpolarity and the lower the adhesion of water molecules on the surface, thus forming a superhydrophobic surface. An increasing number of researchers recognize this view. In order to reduce the surface free energy, it is usually completed by self-adsorption of organic matter in the air or artificial surface modification [35,36,37,38]. Yang et al. [30] exposed the laser-treated surface to the air environment. The surface absorbs oil molecules and organic molecules in the air to produce adsorption reactions, such as formic acid and acetic acid [39], and polymer [40], which reduces the surface free energy, improves the degree of nonpolarity of the surface, and makes it difficult for water droplets to adhere, as shown in Figure 7. They also found that compared with a simple superhydrophobic surface, the superhydrophobic–superhydrophilic composite interface can also inhibit the splash of water droplets and improve the polymerization state of water droplets.

The time of carboxylic acid reaction is consistent with the change curve of the contact angle. The generated carboxylate compound is nonpolar. The high concentration of o – c = o functional group also shows that the carboxylic acid reaction is the reason for low surface energy and nonpolarity [40], and the reaction formula is:(6)$$R-COOH+M-OH\to R-COOM+H_{2}O$$

It is worth noting that a single surface chemical composition or surface morphology cannot make the surface superhydrophobic, but the combination of the two is required [40]. On this basis, the type and number of functional groups will affect the hydrophobicity of the surface [41]. Table 3 outlines common surface modifiers. In order to prove the effect of modifiers on the preparation of superhydrophobic surfaces, Saleema et al. [42] prepared superhydrophobic surfaces only by surface modification.

By changing the thermal factor, the adsorption and decomposition reaction of organics in the modifier can be controlled, thus affecting the wettability of the surface, and the temperature-controlled superhydrophobic surface can be prepared, which has a certain significance in regulating the surface corrosion resistance [49], as shown in Figure 8.

### 2.4. Bionic Slippery Surface

A bionic slippery surface is a type of solid–liquid composite structure formed by injecting low surface energy liquid into micro/nano pores, with the surface friction close to zero. It has excellent liquid repellency, non-adhesion, and self-healing, and has become a research hotspot in the field of surface corrosion resistance, giving it great significance for corrosion resistance, which is explained in detail in Section 4.1. In order to prepare a slippery surface, a three-dimensional porous structure for storing and flowing lubricant needs to be established on the substrate, and then filled with modifier and lubricant [50]. The lubricant stops on its surface under the combined action of surface capillarity, van der Waals force, and gravity [47], and the injected lubricant should be incompatible with external fluid [9], as shown in Figure 9.

Slippery surfaces have better properties than pure superhydrophobic surfaces and are of great importance in corrosion resistance function. For example, common super-hydrophobic surfaces have disadvantages such as low mechanical stability and weak pressure stability, which can be compensated by slippery surfaces [51]. The prepared slippery surface has a certain degree of self-healing [52], as shown in Figure 10, which can improve the degree of corrosion resistance of the surface to a certain extent. Yang et al. [9] validated this statement, emphasizing that coating thickness also affects the strength of surface corrosion protection.

Lubrication performance is mainly affected by lubricant performance. Therefore, controlling the type of lubricant and specific external conditions can adjust the strength of the sliding performance, to prepare a controllable super sliding surface and affect the movement of surface droplets, to further control the strength of corrosion resistance performance. Table 4 shows the common manufacturing of controllable slippery surfaces. 

By using these control techniques, adjustable corrosion-resistant surfaces with different corrosives can be produced on the same slippery surface, which is of great significance for further study of the corrosion protection function. In addition, the lubricant film caused by the “masking effect” (thin lubricating film on droplets hits the surface) should be reduced or avoided [60,61,62], which provides an important reference for building long-term corrosion prevention efficiency on the substrate surface.

## 3. Laser Fabricating Processing and Surface Structure Processing

### 3.1. Selection and Influence of Laser Types

Since a superhydrophobic surface has high corrosion resistance, as described in Section 4.1, the preparation of better micro and nano structures can increase the content and distribution of air cushions, which has a significant impact on the improvement of corrosion resistance. Therefore, it is very important for researchers to improve the laser texturing result by following processing laws. In order to obtain a superhydrophobic surface with corrosion protection, the groove structure that improves the surface roughness of the metal substrate and increases the surface to store gas or liquid is the basis. Laser processing has the advantages of low cost, good stability, moderate preparation period, and low pollution. The schematic diagram of laser etching processing is shown in Figure 11.

The nature of the laser itself affects the processing results. According to the wavelength of the laser source, there are three common types: infrared laser processor, ultraviolet laser processor, and green laser processor. Some researchers believe that under the same processing conditions, the effect of an infrared laser processor might be better than that of an ultraviolet laser [64]. In addition, since the processing effect of green lasers on non-metallic materials is good and the effect on metal processing is general, a green laser is usually not used to texture the metal substrate. Therefore, an infrared laser is the most widely used by researchers. Common laser pulses in time units are continuous laser, nanosecond laser, picosecond laser, and femtosecond laser, as shown in Figure 12. Continuous lasers are generally not used and materials during nanosecond and picosecond laser processing will resolidify in unprocessed surfaces and grooves, resulting in unclear structures. Femtosecond laser has the advantages of ultra-high peak power density, ultra-fast time resolution, and high focusing ability, which sublimates the material directly and avoids melting and accumulation. Therefore, femtosecond laser can process surface structure more accurately [40]. However, due to the small pulse width and short pulse length of femtosecond laser, the processing time is longer and it cannot be used in large quantities and large areas. However, nanosecond and picosecond processing have faster processing times and lower equipment costs, so they are more suitable for large area and mass processing.

### 3.2. Selection and Influence of Laser Parameters

#### 3.2.1. Laser Physical Parameters

The choice of laser parameters will influence the effect of the machined surface structure, thus affecting the superhydrophobic and corrosion resistance performance. It is also necessary to make a specific analysis of different materials and processing conditions. Laser parameters include laser physical parameters and processing parameters. The physical parameters of laser include spot size, energy density, and the number of pulses. The minimum dimension of the microstructures on the ablated surface is determined by the size of the spot, and spot radius is:(7)$$\omega_{0}=\lambda f/(\pi \omega )$$
where $\omega_{0}$ is the spot diameter, λ is the laser wavelength, f is the focal length of the lens, and ω is the spot diameter on the lens surface. The processing accuracy of small spot is higher, but the processing time is long and the efficiency is not high. Energy density is another important parameter of laser itself. Due to the Gaussian distribution of laser beam energy, the processing will produce grooves that gradually become shallower from the middle to the outside. Adjusting the energy density parameter can change the depth and size of the processed groove. When the energy density is increased, the pit depth is deepened and the radius becomes larger, as shown in Figure 13.

Expressed by single pulse energy, the peak fluence in the beam $I_{0}$ is [65]:(8)$$I_{0}=\frac{2E}{\pi \omega_{1}^{2}}$$
where $\omega_{1}$ is the Gaussian beam radius and E is the pulse energy. With the increase in radius, the spatial fluence profile at different radii meets the following relationship, where I(r) is [66]:(9)$$I(r)=I_{0}e^{-\frac{2r_{0}{}^{2}}{\omega_{1}^{2}}}$$
where $r_{0}$ is the distance from the beam centre. From the above formula, it can be calculated that the ablated crater diameter $D_{ab}$ is [67]:(10)$$D_{ab}=\sqrt{2\omega_{1}^{2}\ln E+2\omega_{1}^{2}\ln\left(\frac{2}{\pi \omega_{1}^{2}I_{th}}\right)}$$
where $I_{th}$ is the ablation threshold fluence.

The number of laser pulses N refers to the number of repeated pulses in one second, which is:(11)$$N=\frac{\pi \omega_{0}f_{0}}{4v}$$
where $f_{0}$ is the laser repetition rate and v is the laser scanning speed. The number of pulses is too low, and the fabricating degree in the unit groove is small, which cannot be well processed, resulting in small depth. If the pulse number is too high, the melted quantity will increase, and the liquid metal will accumulate and re-solidify at the bottom of the groove, resulting in unclear groove structure and reduced groove depth. Therefore, the number of pulses per unit groove will affect the depth and structure of the groove structure [64]. In order to obtain a better superhydrophobic effect, the parameter of pulse times should be reasonably selected, as shown in Figure 14.

It is worth noting that when the energy density and the number of pulses are reasonably selected, periodic nano-ripples and nano-particles can be constructed on the material surface, which is of great significance to the formation of a superhydrophobic surface and the construction of the corrosion resistance function. With the increase of energy density, the nano-ripples and nano-particles will increase [68]. Yang et al. [1] adjusted laser physical parameters and prepared periodic nano-ripples and nano-particles on the surface, as shown in Figure 15.

#### 3.2.2. Laser Processing Parameters

Laser processing parameters include scanning speed, scanning distance, spot overlap rate, average power, etc. The scanning speed affects the number of pulses irradiated by the laser on each groove in a unit time. High scanning speed will result in insufficient processing depth, while low scanning speed will lead to molten material entering the groove and solidifying again, thus shallowing the groove first processed. Yang et al. [64] set up six groups of experiments with different scanning speeds. The surface morphology under scanning electron microscope is shown in Figure 16. By measurement, the roughness decreases first and then increases with the increase in scanning distance, with the surface contact angle with moderate scanning speed being the largest. Analysis has discovered two reasons for this phenomenon: one is from the scratch of corrugated structure and hierarchical structure, while the other is from strength to weakness of the spot overlap effect. Due to the thermal expansion and contraction effect, many small cracks and particles will appear at the bottom of the groove. The bigger the scanning speed, the more obvious the cracks and particles will be.

In addition, properly controlling the scanning spacing can induce good nanoparticles. Zhang et al. [47] controlled the scanning spacing to induce nanoparticles on the micro groove and micro prism structure. The researchers also concluded that when the scanning spacing is greater than the beam diameter, the smaller the scanning spacing, the more particles will be generated, as shown in Figure 17. When the scanning distance is smaller than the beam diameter, the scanning path overlaps, resulting in an increase in nanoparticles; however, the microstructure is damaged, resulting in a decrease of hydrophobicity, as shown in Figure 18. Other studies have also shown that the increase in nanoparticles will increase the surface roughness [69].

The average power mainly affects the splash of materials. At low average power, the laser irradiation is weak, resulting in only slight changes in surface morphology, less edge stacking, and smaller groove width and depth. When the average power increases, the stacking volume, width, and depth increase. When the average power increases to a certain value, high temperature will produce a large amount of splashed molten matter, which will splash to the untreated surface area and solidify again. The higher the power, the greater the degree of splashing, and it will even almost cover the raw surface, making the surface rougher. However, the higher the power, the better the structure. When the power is great, the molten material will move slightly before resolidification, and more materials will melt at the groove, to cover and fill the groove, so that the surface morphology is not obvious and the roughness is reduced [64]. By reasonably controlling the above laser parameters, the best corrosion resistance surface can be prepared for different substrates.

### 3.3. Selection and Influence of Processing Medium

Common metal substrates materials include copper alloy [70,71], aluminum alloy [1,30], titanium alloy [72,73,74], ferroalloy [75,76], magnesium alloy [48,77,78], etc. In the processing stage, the material between the laser processor and the metal substrate is called the processing medium. Due to the differences in the properties of the medium itself, it will affect the energy distribution at the processing place and the physical changes of the material, and will then affect the structure of the surface morphology, thus affecting the hydrophobicity and corrosion resistance. Common processing environments include vacuum, gas medium, and liquid medium. In the vacuum environment, because there is no thermal conductive medium, the heat at the processing place will not escape and will be absorbed by the base material, to obtain a clear surface structure. When the substrate is in vacuum and the laser irradiates the carbon layer on the surface, micro nano junctions will be formed. These structures have better adsorption of low free energy substances in a vacuum and shorten the preparation cycle [1]. However, the realization of vacuum conditions is more complex, so it is less applied. The gas medium is the most commonly used medium in the processing environment. Pou et al. [79] compared the effects of processing in five gases and found that a large number of oxide layer cracks appeared on the surface processed in oxygen, and there were periodic microstructure ripples in the groove structure. However, in argon, cracks and corrugated structures will not appear in the grooves, and the structure is clearer. The characterization experiment shows that since there are few polar substances in argon, there is more space for organic substances to be adsorbed on the surface during post-treatment, resulting in a large contact angle in argon, as shown in Figure 19. In addition, high carbon dioxide environment can also inhibit the change from wettability to super hydrophobicity [80]. Air is the most complex processing environment, but it is also the most widely used processing atmosphere.

The liquid medium has been paid increasing attention because of its inhibition to local explosion of base materials. Comparing the processing results in air and water, it was found that the surface structure quality and characterization result data processed in water are better than those in air [81]. Use ethanol solution as the treatment medium. As the liquid layer around the material evaporates during processing, cavitation bubbles will be generated, thus forming a special porous structure on the surface [82]. In addition, the bubbles can also rearrange the molten material on the surface to form a special structure [83]. Organic solution as the processing medium is another important method. Yang et al. [1] used FAS and ethanol as the processing medium. In addition to the cooling effect and the special structure brought by cavitation bubbles, the chemical reaction during irradiation accelerated the adsorption process of nonpolar functional groups, thus shortening the transformation cycle and improving the processing effect, as shown in Figure 20.

### 3.4. Selection and Influence of Processing Structure

There are a large number of hydrophobic surfaces in nature. The mastoid structure on the surface of lotus leaves forms an air cushion, which makes the surface achieve stable superhydrophobic characteristics under the simultaneous action of wax on the epidermis [10]. There are a large number of small hairs in the same direction on the legs of the water strider, forming a spiral nano-groove structure, which can suck air into the groove to form an air bag, preventing water from wetting the infrastructure and forming a superhydrophobic structure [11]. Stratakis et al. [84] performed a detailed research and summary of laser processing based on bionic structure.

Geometric structure is abstracted from these superhydrophobic microscopic electron microscope structures in nature, including lattice structure, groove structure, grid structure, cubic prism, cubic pyramid, mastoid structure, etc., as shown in Figure 21. Among them, grid structure and cubic prism are the most used by researchers. For example, Zhang et al. [47] constructed a cubic prism structure on the surface of brass and prepared a superhydrophobic corrosion resistance functional surface. In addition, Yang et al. [64] compared the hydrophobicity of three different structural surfaces. Due to its large ablation degree and complex surface morphology, the grid structure has the largest roughness. The linear structure ranks second, the point structure is the smallest, and the contact angle of the grid structure is the largest, as shown in Figure 22. By choosing a suitable structure, it is easier to obtain a superhydrophobic surface with good corrosion resistance.

## 4. Corrosion Resistance Function

### 4.1. Corrosion Resistance Mechanism

The global economic loss caused by metal corrosion is about 400 million dollars per year [85]. The chemical or electrochemical reaction between the metal substrate and the environment leads to the destruction and dissolution of the metal [86,87], reducing its service life, such as the corrosion of magnesium alloys [88]. Due to the advantages of good economy, high processing efficiency, convenient processing, large processing scale, low pollution, and strong controllability, laser texture processing is increasingly widely used. In recent years, the use of laser ablation technology has been a major breakthrough in corrosion prevention [89]. After the superhydrophobic surface was made, it was found that the corrosion resistance of the substrate was enhanced compared with the original surface in the acid, alkali, and salt environment under certain specific conditions. Therefore, scientists have conducted in-depth research on this phenomenon, explored the reasons for the enhanced corrosion resistance of the superhydrophobic surface, and improved the corrosion resistance. There are many reasons for the formation of this corrosion resistance effect. First, the surface structure is a big factor. Nanoscale structures on superhydrophobic surfaces can trap gases and form cavitation, which isolates the contact between corrosive substances and the substrate to a certain extent, thus protecting the superhydrophobic surface and achieving a corrosion inhibition effect [1,90,91]. The corrosion resistance of the superhydrophobic surface prepared purely by laser alone has been greatly improved compared with the original surface [35,92]. Mohammed et al. [14] proposed a corrosion resistance model based on the cavitation structure, as shown in Figure 23. The composite interface reduces the contact area between the metal surface and corrosive salt solution to the greatest extent, thus preventing the corrosion of corrosive ions on the metal surface and enhancing the corrosion resistance of the surface.

In addition to the cavitation structure, the corrosion resistance is also related to the resistance of the surface-modified coating to the charge transfer. Through hydrolysis reaction and condensation reaction, the micro molecules of the modifier were self-assembled on the surface of brass and the modified coating was prepared. This coating increases the difficulty of the diffusion of corrosive droplets on the surface, so that corrosive substances can contact the substrate as little as possible, thereby improving the corrosion resistance [47], as shown in Figure 24, which further confirms the role of modified coating in improving the corrosion resistance of superhydrophobic surfaces. Maintaining the long-term effect of modified coatings in corrosive liquids is a major research focus. Li et al. [70] immersed the copper alloy in the modifier to make the organic molecules more closely bound to the surface. This improves the thickness and durability of the superhydrophobic coating, so as to avoid strong oxidation after soaking in alkaline solution for 4 h and improve the long-term corrosion resistance. Zhang et al. [93] prepared a double smart coating with superhydrophobic and self-healing characteristics, in which the superhydrophobic effect of the coating provides the first barrier for the substrate. At present, the most commonly used corrosion-resistant coatings for preparing superhydrophobic surfaces are organic [94] and polymer coatings [95], but the long-term stability of these two coatings in corrosive liquids is weak. The recently developed ceramic coating [96] is a new coating with certain hydrophobic capacity and stability. However, the pores and cracks on the surface of the coating will lead to direct contact and pollution between the substrate and corrosive substances, resulting in the decline of corrosion resistance. Therefore, the type and nature of the coating should be reasonably selected for different substrates. Additional charge transfer barriers are formed from oxide-rich patterned metal surfaces [97]. Boinovich et al. [98] also noted that a superhydrophobic and surface oxide layer have a synergistic effect on corrosion resistance function. Although pure superhydrophobic surfaces have good corrosion resistance, the controllability of the micro roughness, microstructure size, and shape of such surfaces is poor, which hinders the practical application of the corrosion controllability of superhydrophobic surfaces, which is a research direction that needs further improvement [14].

Surface inhomogeneity is also the main factor affecting corrosion resistance. The laser-textured, titanium-based superhydrophobic surface and the untreated surface have been tested for corrosion, and the degree of corrosion degree of the former is significantly lower. The analysis shows that the laser melting surface makes the surface of the substrate uniform, while the nitrogen element penetrates from the surface of the substrate, resulting in surface hardening, thickening the surface hardening layer, and improving its corrosion resistance [33].Bram et al. [99] found that the microstructures of coatings became uniform after heat treatment for 2 h, which greatly enhanced the corrosion resistance, as shown in Figure 25. As for the influence of surface uniformity on corrosion prevention, Shabalovskaya et al. [100] found that the more uneven the substrate surface is, the more vulnerable it is to corrosion. Therefore, eliminating surface inhomogeneity is also one important factor to improve the corrosion resistance of metals.

Some researchers explained the reason for the improvement of corrosion properties from the point of view of crystal structure changes. Song et al. [101] used magnesium and magnesium alloys as experimental objects and noted that an important reason for the increase in corrosion resistance was the formation of heavy grains and secondary phase lattices in the laser ablation area, with many crack-free and defect-free structures produced on the near and sub-surface. The new secondary phase forms more grain boundaries, which decreases the activity of the corrosive microcell as the grain boundaries increase, while the continuous distribution of the secondary phase further protects the surface from corrosion and enhances the corrosion resistance, as shown in Figure 26. In addition, Ballerini et al. [77] noted that the corrosion protection and the content of some enriched phases are proportional, and the corrosion resistance of the base material depends on many factors, such as composition, impurity element content and composition, processing technology, and crystal structure.

Researchers found that if the superhydrophobic surface is prepared into a slippery surface, its corrosion resistance will be greatly improved. Therefore, the preparation of a slippery surface to enhance the corrosion resistance of the surface has been adopted by more researchers. After the slippery characteristic is added to the super hydrophobic surface, the surface produces a “self-healing” characteristic through the inherent photothermal effect, the reversible coordination of the chemical structural blocks of the matrix main chain, and the diffusion and synergy of the lubricating fluid, which is very helpful in improving the corrosion resistance [51]. On the one hand, lubricants further weaken the retention capacity of corrosive liquids on the surface. On the other hand, the multi-layer pore structure provides space for the flow of lubricant. When the surface is subject to chemical corrosion or mechanical damage, this structure can allow the lubricant to flow inside to repair the damaged parts. If lubricants with good heat resistance and low volatility are used, the corrosion resistance and durability of the surface can be further improved [52]. Yang et al. [9] demonstrated this by comparing the corrosion resistance of ordinary superhydrophobic surfaces and slippery surfaces, the latter performs better.In addition to the instantaneous corrosion resistance, they also found that the slippery surface formed a special lubrication barrier due to the injection of surface modifier and silicone oil. When pitting occurs on the surface, it can recover in a short time to prevent the large-scale diffusion of corrosion, so it has better long-term corrosion resistance than ordinary superhydrophobic surfaces. In addition, the lubrication film caused by the “masking effect” (droplets hitting the thin lubrication film on the surface) should be reduced or avoided [60,61,62], which provides an important reference for establishing long-term anticorrosive efficiency on the substrate surface.

It is noteworthy that the corrosion resistance stability of the surface prepared by this process needs to be improved. At present, some surfaces can still maintain corrosion resistance after soaking in corrosive solution for 4 h, but there is still room for improvement [70]. In summarizing the research on corrosion resistance, the corrosion resistance stability can be improved from the perspective of the mechanism. Due to the isolation effect of air cushions in the surface microstructure, researchers should explore better laser parameters to obtain more air cushions through the law between laser parameters and surface quality. After laser texturing, the use of surface modifiers with better ion isolation is also one method to enhance stability. If the surface is filled with lubricant to make a slippery surface, the pitting effect will be weakened and the corrosion resistance and stability will be further improved. Reducing the inhomogeneity of the surface and improving the crystal structure of the surface are the microscopic approaches.

### 4.2. Corrosion Resistance Characterization Experiments

#### 4.2.1. CA/RA Experiment

The contact angle is an important experiment to measure the superhydrophobic property of a surface. A contact angle tester is generally used to measure the contact angle [16]. A drop of liquid is dripped through the dropper and the droplet is attached to the measuring surface by the operating platform. After the droplet is stabilized, the contact angle is shown in Figure 27. In addition, the rolling angle measurement can be based on the contact angle-measuring instrument with a tilting device to record the critical angle of the droplet roll as the rolling angle. In order to reduce measurement errors, a method of five-point sampling and averaging a sample surface is provided, which improves the objectivity and accuracy of data [47]. The hydrophobicity of the surface obtained under different processing conditions will vary. When the surface has a superhydrophobic capacity, the droplets will stand on the surface with a contact angle greater than 150° [102], as shown in Figure 27.

#### 4.2.2. Scanning Electron Microscope Experiments (SEM)

Scanning electron microscopy (SEM) is a means of observing micro-topographic surfaces [103]. For prepared superhydrophobic surfaces, researchers want to observe micro-scale surface topographic structures such as micro groove structures [104] in the SEM image, as shown in Figure 28.

In addition to the microstructure, periodic nanoparticles need to be observed by SEM experiments. In the preparation of superhydrophobic surfaces, nanoparticles are representative structures in nano-size. Under the action of laser-induced ablation and explosion, the bulges generated by ablation and explosion are splashed and cooled, resulting in the formation of nano-spherical particles and needle-like bulges on the surface [105], as shown in Figure 29.

#### 4.2.3. X-ray Photoelectron Spectroscopy Experiment (XPS)

X-ray photoelectron spectroscopy (XPS) can determine the type and content of chemical bonds on the surface. The type and content of chemical bonds can be calculated by electron peak offset to determine the indicated calculated molecular structure. The effect of laser treatment and surface modification on superhydrophobic and corrosion resistance can be analyzed and the influence of a functional group on properties can be seen. For example, Liu et al. [105] found that photoelectron spectroscopy experiments on prepared superhydrophobic surfaces showed an increase in silicon and carbon content, indicating that alkyl–silicon films with low surface free energy have wrapped the substrate surface, enhancing the hydrophobic and corrosion resistance properties of the substrate, as shown in Figure 30.

#### 4.2.4. Energy Dispersion Spectrum Experiment (EDS)

According to the energy spectrum experiment, the percentage of alloy elements and impurities occupying a sub-lattice position in the compound can be determined, the type and content of surface elements can be visually seen, the influence of processing and corrosion experiments on the surface can be analyzed, and the corrosion resistance and superhydrophobic properties of the surface can be judged. For example, the surface fluorine content decreases with an increase in corrosion time. The effect of surface modification on superhydrophobicity and corrosion resistance as well as its stability is explained in detail [9], as shown in Figure 31.

#### 4.2.5. Dynamic Potential Polarization Curve Experiment (PDP)

The dynamic potential polarization curve test is an experimental method used to detect the corrosion resistance of metals. The Tafel curve is plotted by recording potential deviation generated by electrolysis of the measured surface to determine the strength of the corrosion resistance. The corrosion potential shows the difficulty of corrosion. The greater the positive shift of corrosion potential and the smaller the corrosion current, the stronger the corrosion resistance. Zhang et al. [47] carried out PDP experiments on three surfaces, as shown in Figure 32. From the results, the corrosion resistance of superhydrophobic and slippery surfaces was the strongest, followed by superhydrophobic surfaces, pure slippery surface ranked the third, and pure surface ranked the worst.

#### 4.2.6. Electrochemical Impedance Spectroscopy Experiment (EIS)

Electrochemical impedance spectroscopy (EIS) is used to analyze the degree and cause of electrochemical corrosion by measuring the changes of impedance and corrosion current. See formula 12 to calculate impedance values. Another method is to observe the impedance phase angle as an indicator [9].
(12)$$R_{p}=\frac{\beta_{a}\times \beta_{c}}{2.303I_{corr}(\beta_{a}+\beta_{c})}$$
where $R_{p}$ is the polarization resistance and $\beta_{a}$ and $\beta_{c}$ are the Tafel slopes of the anode diagram and cathode diagram, respectively. The corresponding equivalent circuit diagram and Nyquist and Bode diagrams can be drawn by calculation and analysis. The corrosion resistance of the surface is positively correlated with the OCP value, the diameter of the Nyquist ring, the impedance modulus of the low frequency band, the phase angle in the Bode diagram, and the equivalent resistance in the equivalent circuit, but negatively correlated with the equivalent capacitance, as shown in Figure 33 and Figure 34. It shows that the surface processed by laser texture and modified has better corrosion resistance.

## 5. Conclusions and Outlook

Corrosion is a major problem with metal substrates, with electrochemical corrosion the more common form. Bioinspired special wetting surfaces prepared by laser texturing are resistant to corrosion due to their unique surface structure and oxide layer. On this basis, surface post-treatments such as surface modification and lubricant filling can further enhance their ability and is of great significance in the research of corrosion resistance. The superhydrophobic surface and slippery surface are two common types of bioinspired special wetting surfaces. Therefore, based on laser texturing technology, this paper summarizes the preparation of superhydrophobic surfaces and slippery surfaces, including the mechanism of structure preparation and post-treatment. Secondly, the properties of superhydrophobic surfaces are closely related to the processing process, so the laws of laser texturing are analyzed, including laser types, laser parameters, processing media, and texturing structures. The corrosion resistance mechanism found at present is then summarized, including an air cushion in the microstructure, resistance of surface modification layer to charge transfer, protection of oxide layer, reduction of surface inhomogeneity, change of crystal structure, change of surface composition, and self-healing of a slippery surface. By determining the characterization experiments related to corrosion resistance, this paper provides a clue for researchers to evaluate the corrosion resistance of prepared surfaces.

The application of laser processing technology to pretreat superhydrophobic surfaces has achieved good results in corrosion resistance, but there are still some problems and development space:

(1) Laser has good controllability and operability in surface structure construction. Therefore, in order to prepare better superhydrophobic surfaces, we summarized the theoretical model of hydrophobicity and the influence of laser parameters on the results. At present, researchers widely use Cassie Baxter model as the theoretical basis. Although Herminghaus model has a good structure and is important for preparing superhydrophobic surfaces, few researchers use it. Researchers should increase the use of this model in further researches.

(2) The coating produced by a modifier is an important factor to improve the corrosion resistance of a substrate. The corrosion resistance of the surface is closely related to the performance of the modifier. Therefore, researchers should look for better corrosion resistance modifiers to promote the wide application of this technology in industrial production and daily life.

(3) Some slippery surfaces based on superhydrophobic surfaces have good corrosion resistance, so the preparation of slippery performance is an important method to improve corrosion resistance. The ideal slippery surface needs three-dimensional channel structure and good lubricating oil filling. However, at present, most laser processing can only process two-dimensional planar structures and there is a certain development space for the construction of three-dimensional channels. Therefore, improving the three-dimensional manufacturing quality of a substrate is a major development direction.

(4) A laser with different pulse length has its advantages and disadvantages in the processing process. The melting effect of nanosecond laser processing will cause the surface structure to be unclear, but it has high processing efficiency and can be processed in large areas. Femtosecond laser processing has higher precision and clearer structure, but the processing efficiency is low. Therefore, researchers should improve the accuracy or efficiency of different lasers by setting processing parameters to improve product quality and production efficiency.

(5) In the case of laser texture, processing on the micro scale is random and thus less controllable. In order to control the differences of processing results and produce better microstructure results, some researchers have used liquid media as processing media to produce better surface effects. Researchers should look for better processing media to improve surface structure controllability and surface quality.

(6) At present, surface samples prepared by researchers are usually resistant to corrosion for only a short time, such as instantaneous pitting or soaking. However, the corrosion resistance of long-term immersion in corrosive substances will gradually decrease, resulting in the failure of its practical application in many fields. Therefore, improving the corrosion resistance of superhydrophobic surfaces is an important development direction.

## Figures and Tables

**Figure 1 micromachines-13-01431-f001:**
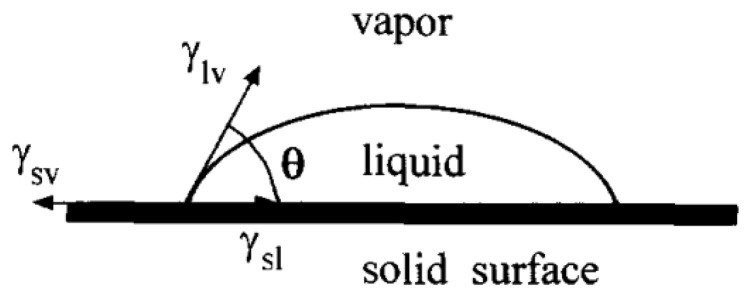
Definition of contact angle (Reprinted with permission from Ref. [16]. 1996, Li, D.Q.).

**Figure 2 micromachines-13-01431-f002:**
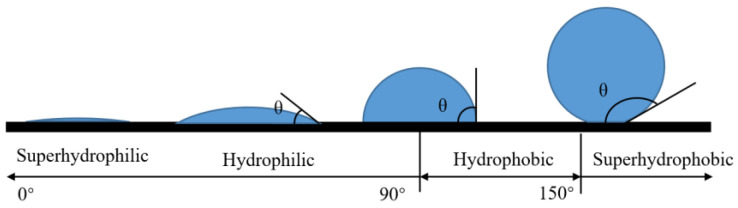
Division of wettability by contact angle.

**Figure 3 micromachines-13-01431-f003:**
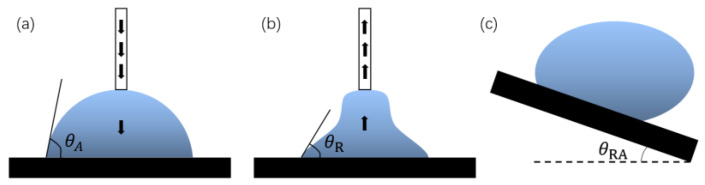
Definition of angles: (**a**) advance angle, (**b**) retraction angle, and (**c**) rolling angle.

**Figure 4 micromachines-13-01431-f004:**
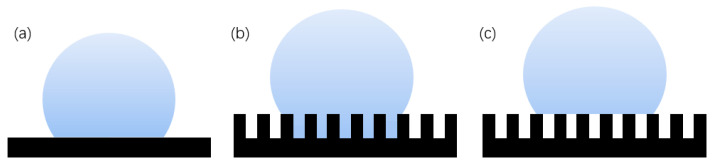
Theoretical model of hydrophobicity: (**a**) Young’s model, (**b**) Wenzel structure model, and (**c**) Cassie–Baxter structure model.

**Figure 5 micromachines-13-01431-f005:**
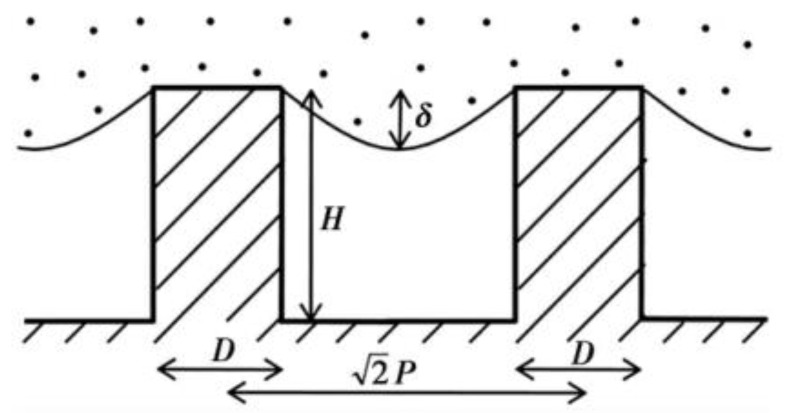
Liquid level and cylinder height of diagonal section of cylindrical structure (Reprinted with permission from Ref. [26]. 2008, Bhushan, B.).

**Figure 6 micromachines-13-01431-f006:**
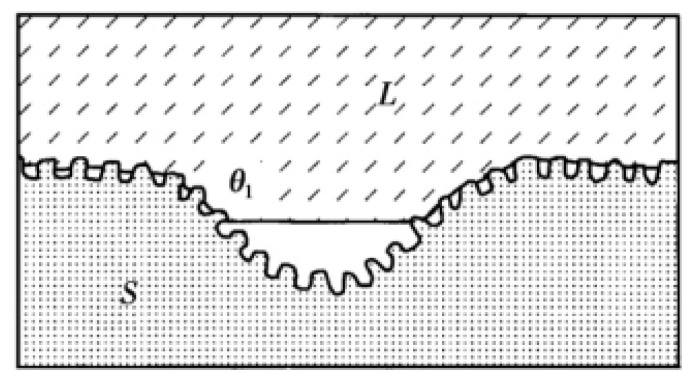
Herminghaus structure model (Reprinted with permission from Ref. [27]. 2000, Herminghaus, S.).

**Figure 7 micromachines-13-01431-f007:**
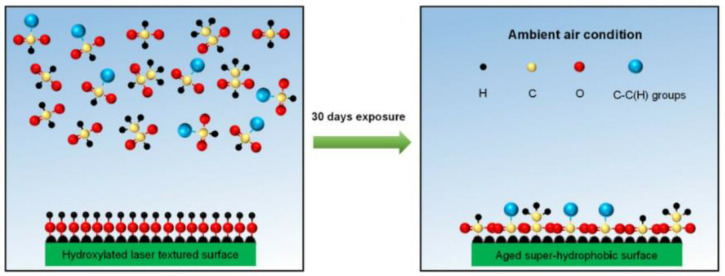
Schematic diagram of chemical adsorption mechanism during placement (Reprinted with permission from Ref. [30]. 2018, Yang, Z.).

**Figure 8 micromachines-13-01431-f008:**
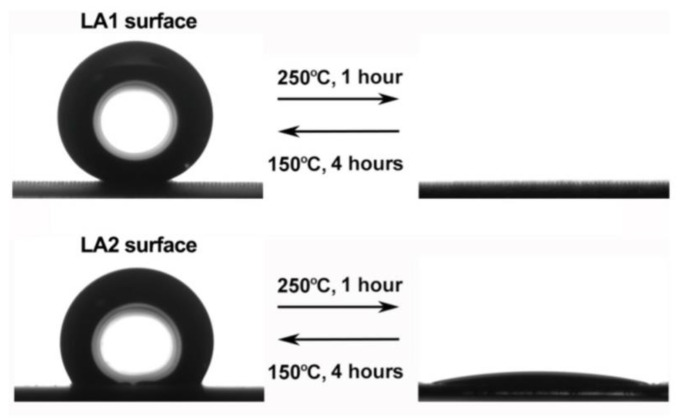
Reversible change of surface wettability by controlling temperature change (Reprinted with permission from Ref. [49]. 2019, Li, X.Y.).

**Figure 9 micromachines-13-01431-f009:**
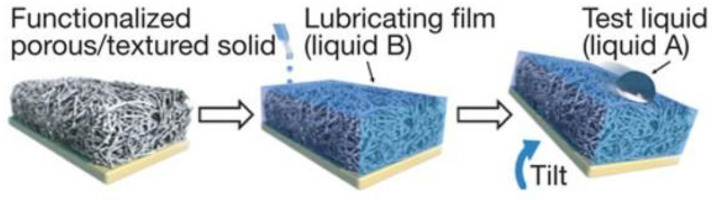
Schematic diagram of preparation process of a slippery surface (Reprinted with permission from Ref. [50]. 2011, Wong, T.S.).

**Figure 10 micromachines-13-01431-f010:**
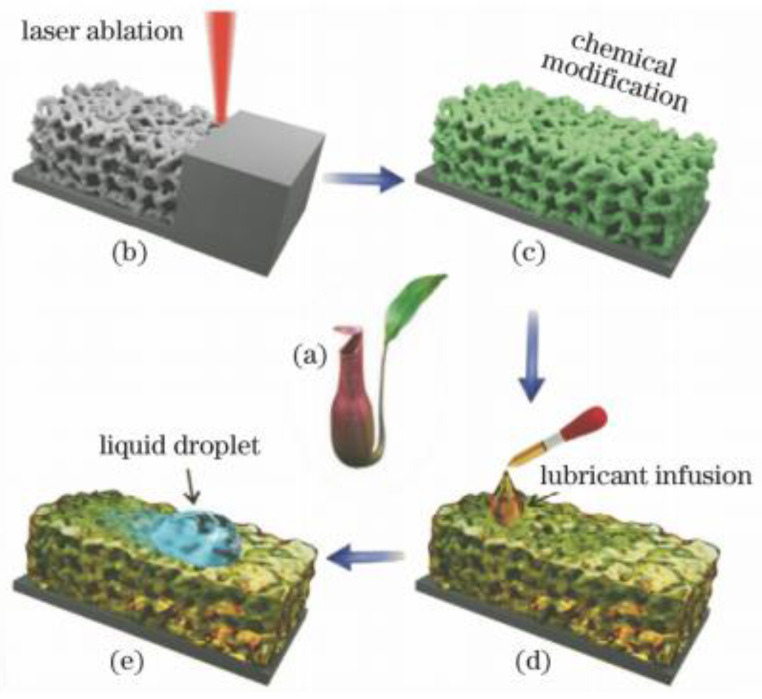
Preparation of a slippery surface of bionic pitcher. (**a**) Photo of the Nepenthes pitcher plant. (**b**) Femtosecond laser ablation used to generate interconnected porous microstructures. (**c**) Fluoroalkylsilane modification used to lower the surface free energy. Green color denotes the fluorosilane molecular layer. (**d**) Infusion of the lubricating liquid (silicone oil) into the laser-induced micropores. (**e**) Foreign liquid droplet sliding down the as prepared slippery surface. (Reprinted with permission from Ref. [52]. 2018, Yong, J.L.).

**Figure 11 micromachines-13-01431-f011:**
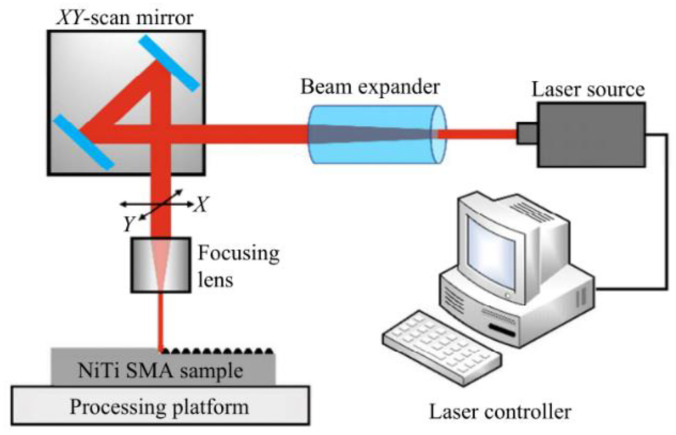
Schematic diagram of laser fabricating (Reprinted with permission from Ref. [63]. 2021, Yang, C.J.).

**Figure 12 micromachines-13-01431-f012:**
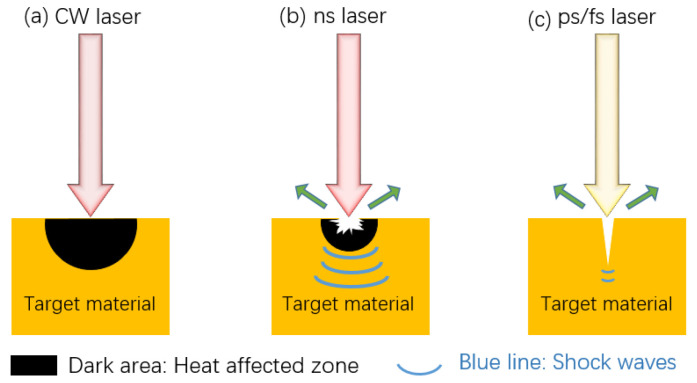
Differences of three types of laser fabricating: (**a**) continuous laser, (**b**) nanosecond laser, and (**c**) picosecond, femtosecond laser.

**Figure 13 micromachines-13-01431-f013:**
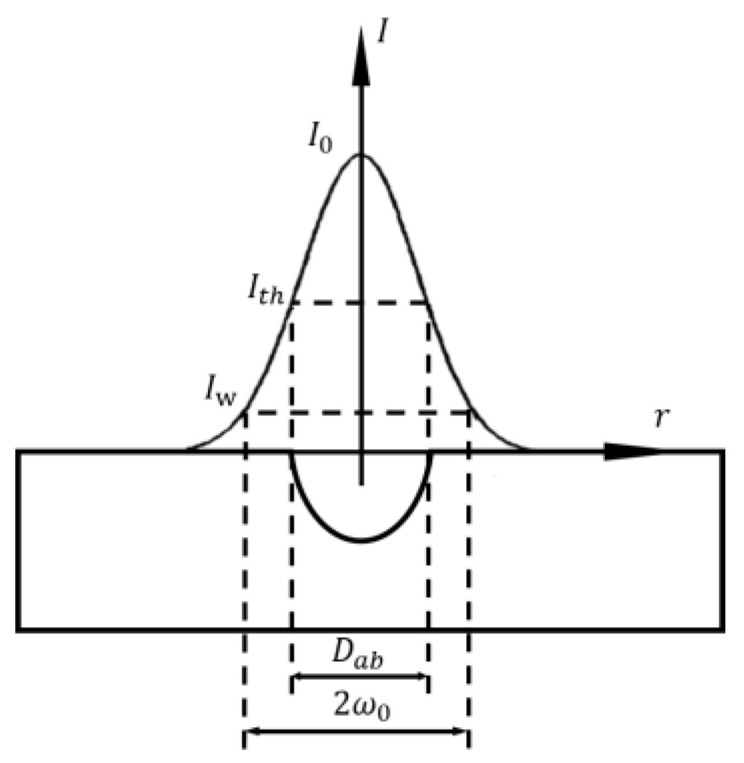
Energy density and fabricating depth (Adapted with permission from Ref. [65]. 2004, Mannion, P.T.).

**Figure 14 micromachines-13-01431-f014:**
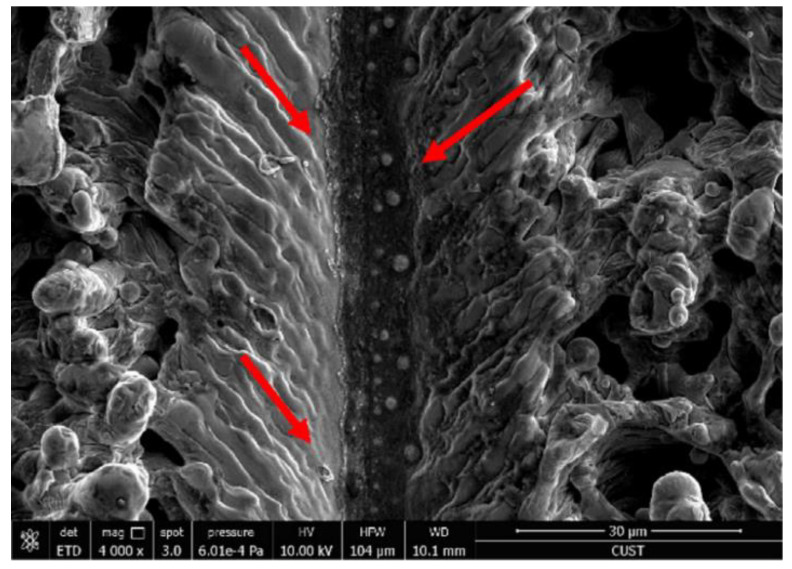
High pulse times lead to recoagulation at the bottom of the groove (molten material flows in the direction of the red arrow) (Reprinted with permission from Ref. [64]. 2017, Yang, Z.).

**Figure 15 micromachines-13-01431-f015:**
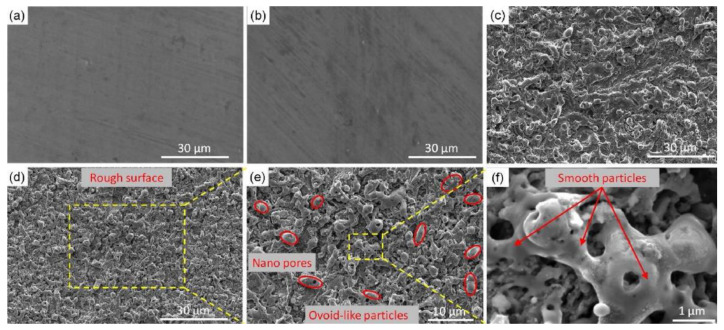
SEM images of (**a**) bare aluminum; (**b**) bare aluminum sample immersed into FAS/ethanol solution for 12 h but in absence of laser treatment; (**c**) bare aluminum sample irradiated into a mixed solution containing distilled water and absolute ethanol but without FAS. The samples immersed in FAS/ethanol solution for 12 h after laser fabrication with scale bar of (**d**) 30 μm, (**e**) 10 μm and (**f**) 1 μm. (Reprinted with permission from Ref. [1]. 2020, Yang, Z.).

**Figure 16 micromachines-13-01431-f016:**
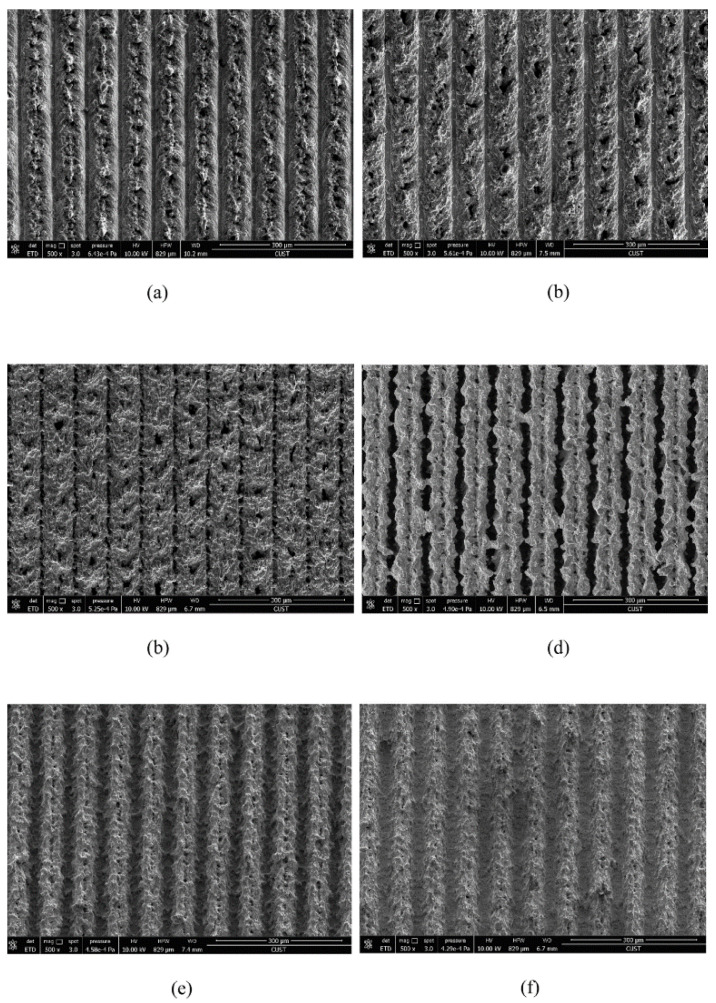
SEM images of surface structure under different laser scanning speeds: (**a**) 20 mm/s, (**b**) 30 mm/s, (**c**) 40 mm/s, (**d**) 50 mm/s, (**e**) 60 mm/s, and (**f**) 70 mm/s (Reprinted with permission from Ref. [64]. 2017, Yang, Z.).

**Figure 17 micromachines-13-01431-f017:**
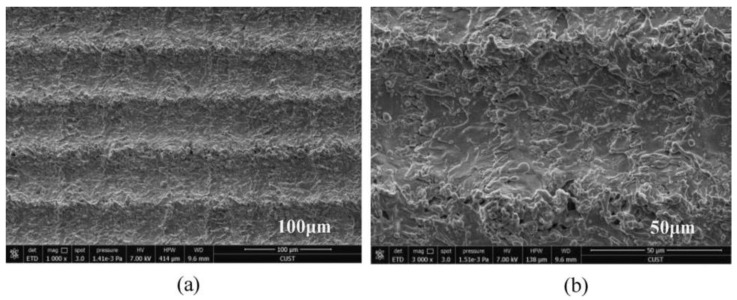
SEM image of laser-induced superhydrophobic surface when the scanning spacing is greater than the beam diameter (**a**) 100 μm length scale (**b**) 50 μm length scale (Reprinted with permission from Ref. [47]. 2021, Zhang, L.).

**Figure 18 micromachines-13-01431-f018:**
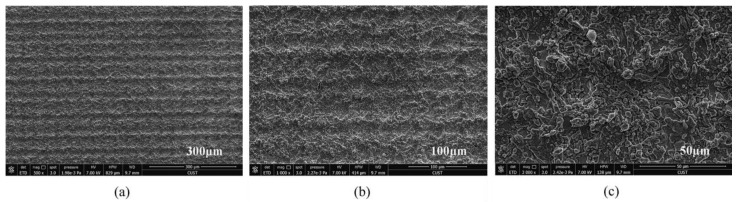
SEM image of laser-induced superhydrophobic surface when the scanning spacing is less than the beam diameter (**a**) 300 μm length scale (**b**) 100 μm length scale (**c**) 50 μm length scale (Reprinted with permission from Ref. [47]. 2021, Zhang, L.).

**Figure 19 micromachines-13-01431-f019:**
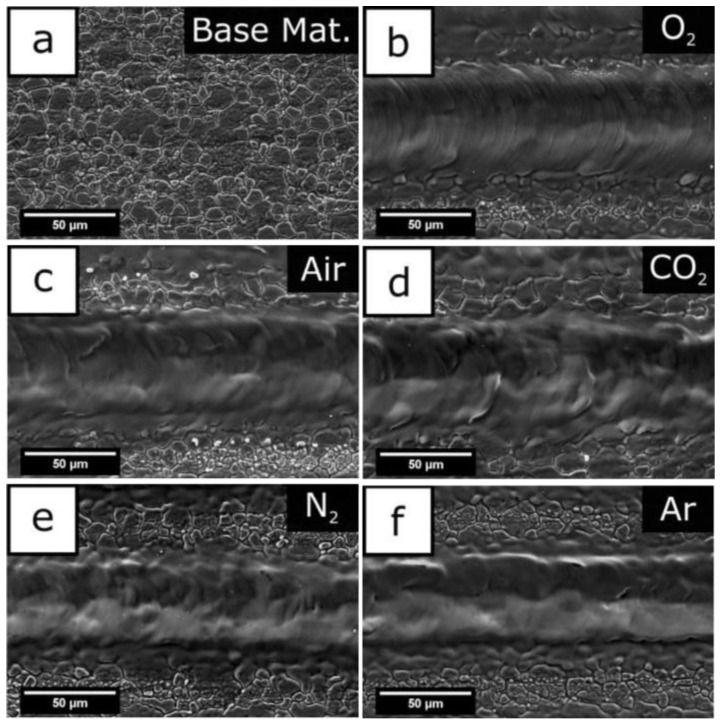
(**a**) Foundation base, (**b**) oxygen atmosphere processing, (**c**) air atmosphere processing, (**d**) carbon dioxide atmosphere processing, (**e**) nitrogen atmosphere processing, and (**f**) argon atmosphere processing (Reprinted with permission from Ref. [79]. 2019, Pou, P.).

**Figure 20 micromachines-13-01431-f020:**
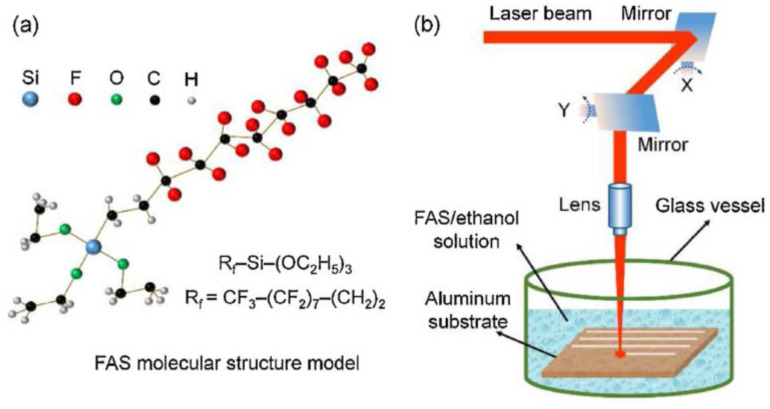
Schematic diagram of processing with ethanol solution of FAS as the processing medium (**a**) molecular structure model of FAS, (**b**) experimental set-up for nanosecond laser treatment in FAS/ethanol solution. (Reprinted with permission from Ref. [1]. 2020, Yang, Z.).

**Figure 21 micromachines-13-01431-f021:**
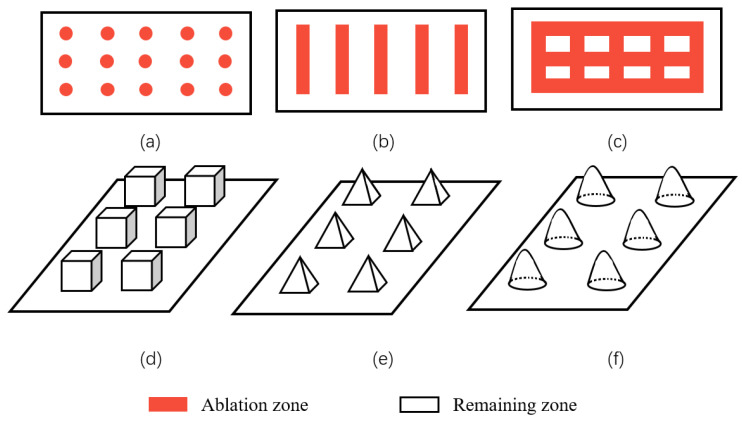
Common surface structures: (**a**) spot structure, (**b**) groove structure, (**c**,**d**) grid and cubic prism structure, (**e**) cubic pyramid structure, and (**f**) mastoid structure.

**Figure 22 micromachines-13-01431-f022:**
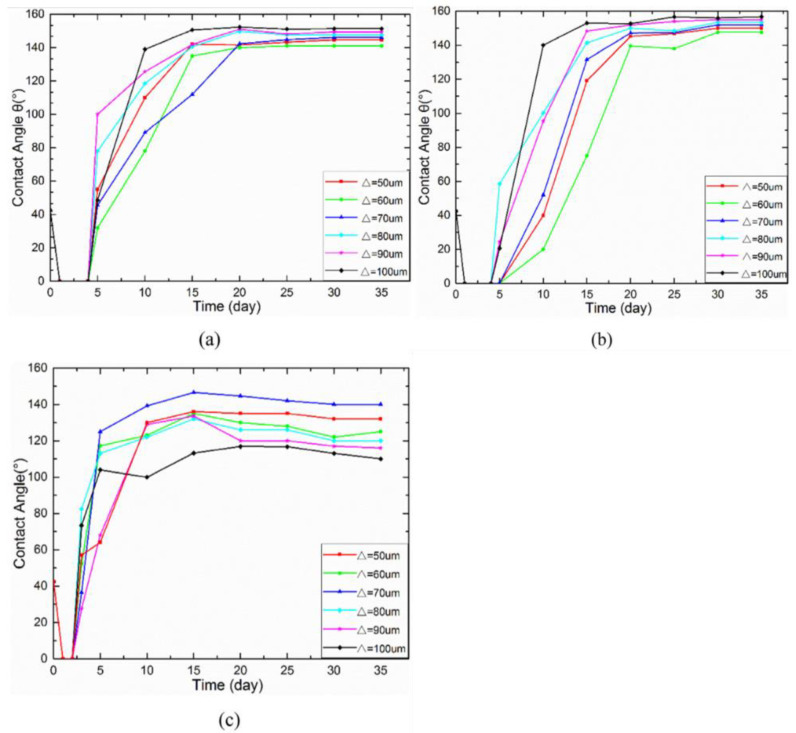
Evolution of the contact angle over time for: (**a**) line pattern, (**b**) grid pattern, and (**c**) spot pattern (Reprinted with permission from Ref. [64]. 2017, Yang, Z.).

**Figure 23 micromachines-13-01431-f023:**
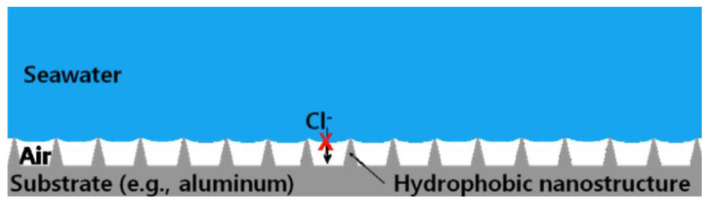
Effect of cavitation structure on corrosion resistance function (The red cross indicates that chloride ions cannot contact the metal substrate) (Reprinted with permission from Ref. [14]. 2015, Mohamed, A.M.A.).

**Figure 24 micromachines-13-01431-f024:**
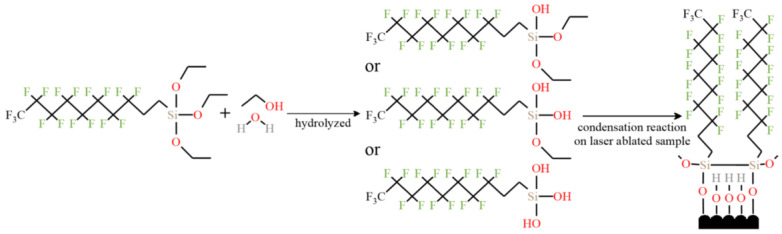
Microscopic reaction of forming superhydrophobic film by self-assembly on a brass surface (Reprinted with permission from Ref. [47]. 2021, Zhang, L.).

**Figure 25 micromachines-13-01431-f025:**
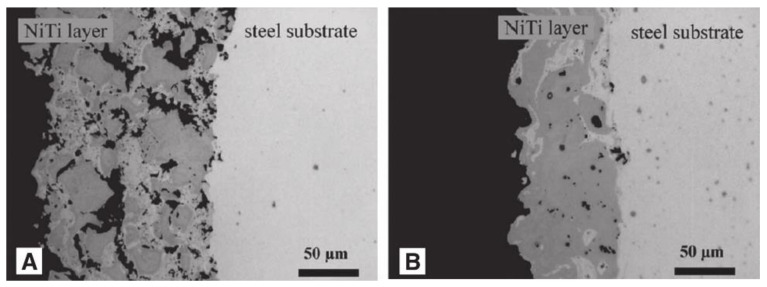
TiNi alloy coatings on a steel substrate produced by VPS. (**A**) The coating is porous and inhomogeneous. (**B**) The coating is less porous and exhibits a higher homogeneity (Reprinted with permission from Ref. [99]. 2002, Bram, M.).

**Figure 26 micromachines-13-01431-f026:**
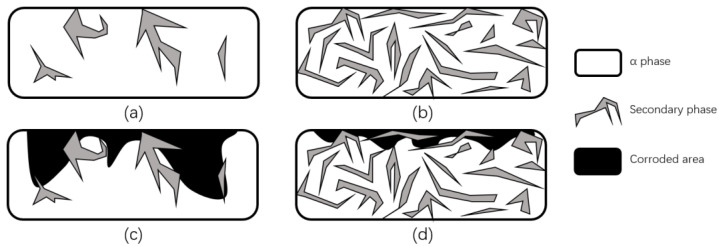
Schematic diagram of corrosion before and after LSM. (**a**) Distribution of a small amount of discontinuous secondary phase in magnesium alloys. (**b**) Distribution of a large number of continuous secondary phases in magnesium alloys. (**c**) Corrosion damage of magnesium alloys containing a small amount of discontinuous secondary phase. (**d**) Corrosion damage of magnesium alloys containing a large number of continuous secondary phases (Adapted with permission from Ref. [101]. 1999, Song, G.L.).

**Figure 27 micromachines-13-01431-f027:**
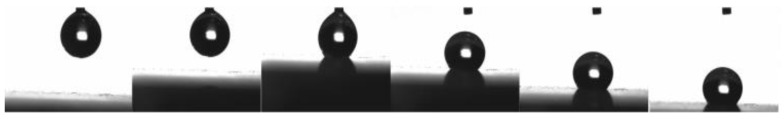
Measurement process of droplet WCA operation (Reprinted with permission from Ref. [69]. 2019, Yang, C.J.).

**Figure 28 micromachines-13-01431-f028:**
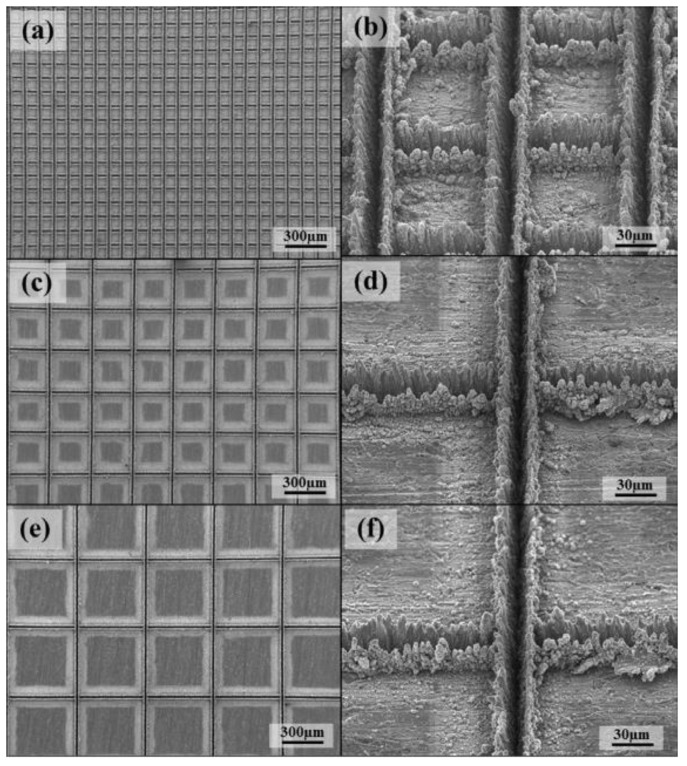
SEM images of laser beam machined surface: (**a**) 100 mm, (**c**) 300 mm, and (**e**) 500 mm, and enlarged images: (**b**) 100 mm, (**d**) 300 mm, and (**f**) 500 mm (Reprinted with permission from Ref. [104]. 2016, Chun, D.M.).

**Figure 29 micromachines-13-01431-f029:**
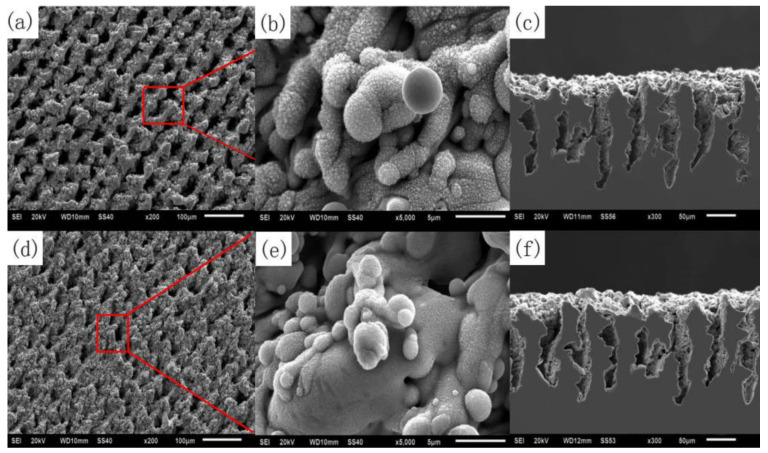
SEM images of laser treated carbon steel before and after coating with PHPS: (**a**) surface microstructure before coating with PHPS; (**b**) surface nanostructure before coating with PHPS; (**c**) cross-sectional morphology before coating with PHPS; (**d**) surface microstructure after coating with PHPS; (**e**) surface nanostructure after coating with PHPS; (**f**) cross-sectional morphology after coating with PHPS. (The red box and line represent the image of the selected area) (Reprinted with permission from Ref. [105]. 2018, Song, J.J.).

**Figure 30 micromachines-13-01431-f030:**
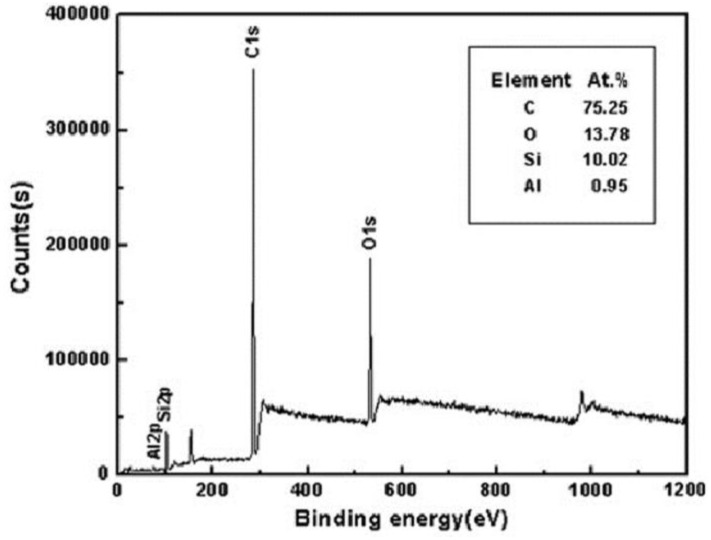
XPS spectral analysis of aluminum alloy superhydrophobic surface by laser treatment and DTS modification (Reprinted with permission from Ref. [106]. 2018, Li, D.W.).

**Figure 31 micromachines-13-01431-f031:**
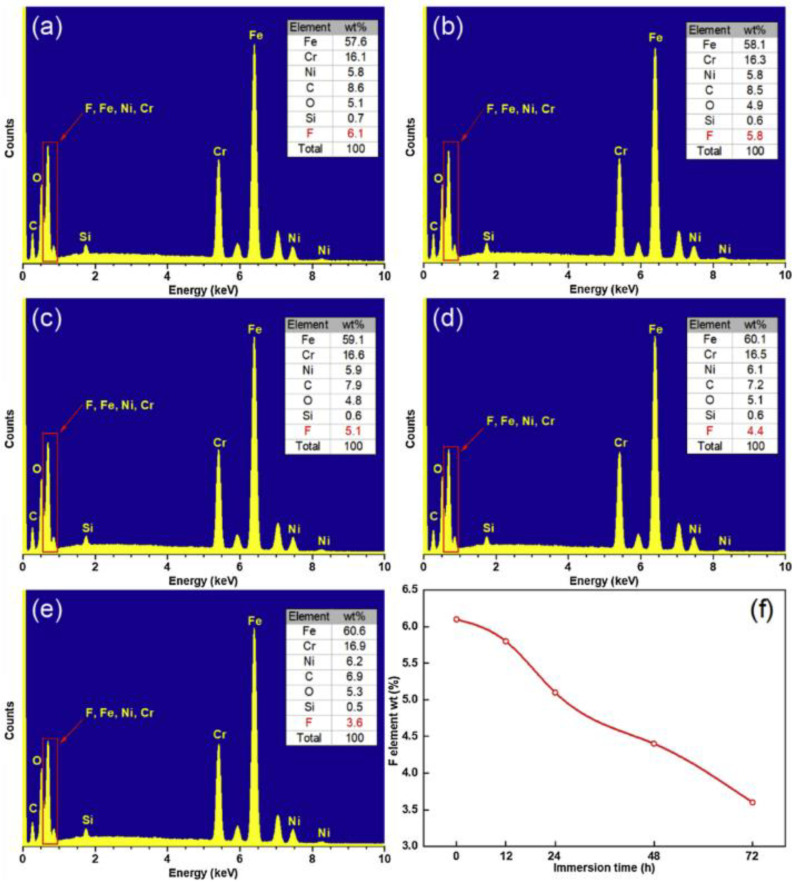
EDS spectrum of superhydrophobic surface prepared after soaking in NaCl solution for: (**a**) 0 h, (**b**) 12 h, (**c**) 24 h, (**d**) 48 h, and (**e**) 72 h. (**f**) The change of fluorine content with soaking time (Reprinted with permission from Ref. [9]. 2020, Yang, Z.).

**Figure 32 micromachines-13-01431-f032:**
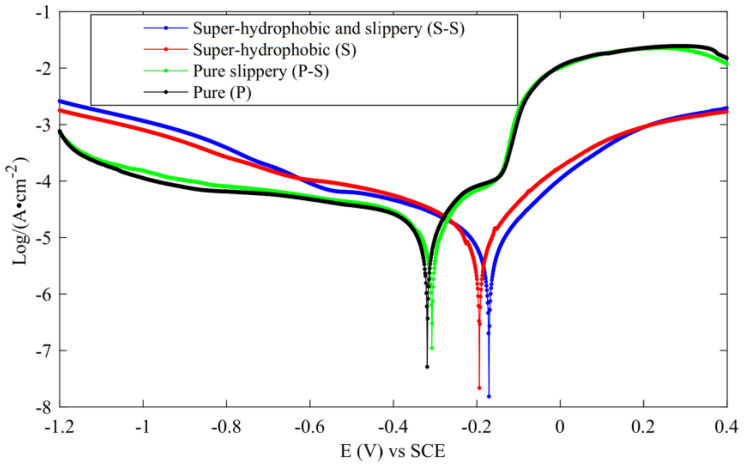
PDP curve (Reprinted with permission from Ref. [47]. 2021, Zhang, L.).

**Figure 33 micromachines-13-01431-f033:**
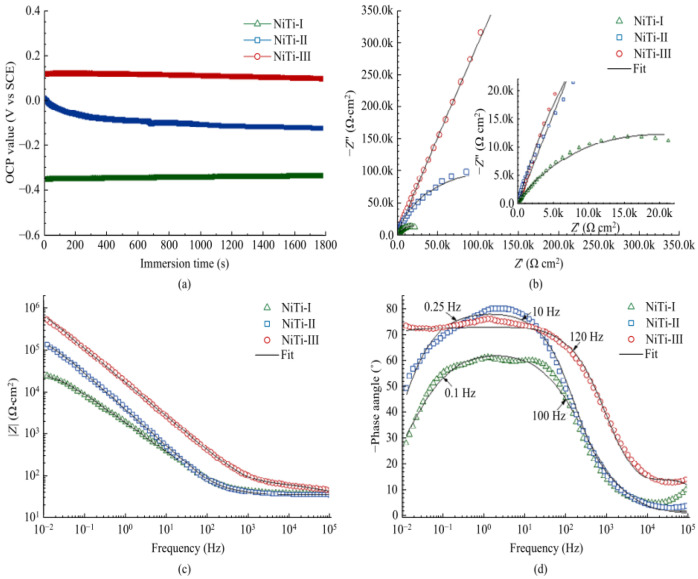
EIS results of NiTi−I, NiTi−II, and NiTi−III samples in SBF: (**a**) OCP curve, (**b**) Nyquist diagram and fitting curve, (**c**) Bode diagram and fitting curve of impedance modulus with frequency, and (**d**) Bode diagram and fitting curve of phase angle with frequency (Reprinted with permission from Ref. [63]. 2021, Yang, C.J.).

**Figure 34 micromachines-13-01431-f034:**
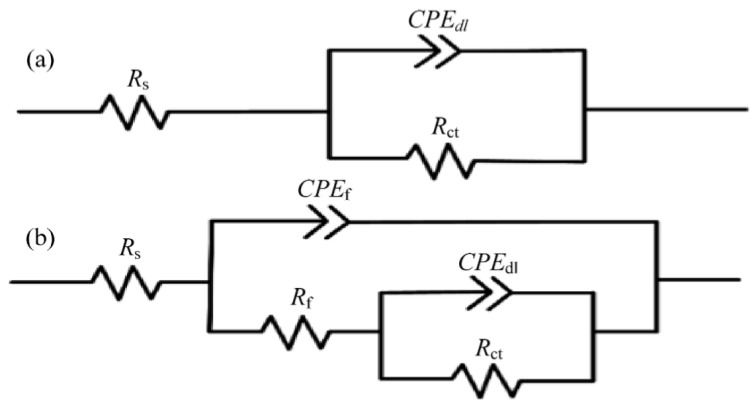
Equivalent circuit models matched to the EIS plots of: (**a**) NiTi−I and (**b**) NiTi−II and NiTi−III samples (Reprinted with permission from Ref. [63]. 2021, Yang, C.J.).

**Table 1 micromachines-13-01431-t001:** Treatment methods of common corrosion resistance.

Preparation Methods	Advantage	Shortcoming	Reference
Laser texturing	Less restrictions and pollution, high efficiency	Interfered by electromagnetic wave	[1]
Electrochemical deposition	High coverage	Internal stress of coating	[2]
Chemical etching	High reliability	High unevenness and danger of etching	[3]
Hot embossing	Good biocompatibility and flexible processing	High temperature and pressure, poor stability	[4]
Chemical vapor deposition	Strong controllability	Inefficient and corrosive	[5]
Sol–gel method	Strong evenness and feasibility	Expensive and inefficient	[6]
Hydrothermal method	Low cost and strong controllability	High temperature conditions	[7]
Spraying method	High accuracy and good uniformity	Shadowing effect	[8]

**Table 2 micromachines-13-01431-t002:** Summary of variables’ representation.

Variables	Representation	Variables	Representation	Variables	Representation
θ	Contact angle	R	Radius of curvature	$D_{ab}$	Ablated crater diameter
$\theta_{A}$	Advance angle	H	Height	$I_{th}$	Ablation threshold fluence
$\theta_{R}$	Retraction angle	n	Series	N	Number of laser pulses
$\theta_{RA}$	Rolling angle	$\theta_{n}$	Apparent contact angle of the nth stage structure	$f_{0}$	Laser repetition rate
$\gamma_{sg}$	Surface tension of solid–gas	$\omega_{n}$	Gas–liquid ratio factor of the nth stage structure	v	Laser scanning speed
$\gamma_{sl}$	Surface tension of solid–liquid	$\omega_{0}$	Spot diameter	$R_{p}$	Polarization resistance
$\gamma_{gl}$	Surface tension of gas–liquid	λ	Laser wavelength	$\beta_{a}$	Tafel slope of anode diagram
r	Surface roughness factor	f	Focal length of the lens	$\beta_{c}$	Tafel slope of cathode diagram
$\theta_{r}$	Apparent contact angle	ω	Spot diameter on the lens surface		
$\theta_{e}$	Intrinsic contact angle	$I_{0}$	Peak fluence in the beam		
$f_{1}$	Solid–liquid ratio factor	$\omega_{1}$	Gaussian beam radius		
$f_{2}$	Gas–liquid ratio factor	E	Pulse energy		
P	Pitch	I(r)	Spatial fluence profile		
D	Diameter	$r_{0}$	Distance from the beam center		

**Table 3 micromachines-13-01431-t003:** Common surface modifiers.

Base Alloy	Modifier	Reference
Aluminum alloy	Perfluoro decyl triethoxysilane, AC-FAS	[1,43]
Stainless steel	Perfluoro decyl triethoxysilane, perfluoro dodecyl trichlorosilane, 1H,1H,2H,2H-perfluorodecyltriisopropoxysilane	[9,40,44]
Titanium alloy	Fluorinated silane, 1H, 1H, 2H, 2H-perfluorodecyltriisopropoxysilane	[12,45]
Copper alloy	Silver nitrate, thioundecanoic acid, dodecyl mercaptan, tetradecanoics acid, perfluoro decyl triethoxysilane	[13,46,47]
Magnesium alloy	Fluorosilane	[48]

**Table 4 micromachines-13-01431-t004:** Controllable slippery surfaces.

Controlling Factor	Lubricant	Performance	Comments	Reference
Temperature	Liquid paraffin	Slippery at high temperatures, not at low temperatures	Specific critical temperature	[53]
Magnetic field	Silicone oil-based magnetic fluid	Slippery without magnetic field, not with magnetic field	Microstructure can be controlled	[54,55]
Electric field	Conductive silicone lubricant	Electric field parameters affect the motion of conductive droplets	Silicone oil viscosity affects self-healing	[56]
Photocatalytically	PDMS lubricant	Be slippery with photocatalytically	Titanium alloy oxide as substrate	[57]
Photothermal effect	Organic gel with ${\mathrm{Fe}}_{3}O_{4}$ particles	Control the motion of surface droplets	Surface droplets include water and alcohol	[58]
Photoelectricity	P3HT/PCBM (interfacial directional freezing)	Control the motion of surface droplets	Affected by freezing rate and mass ratio	[59]

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
