# Peer review of "Research on Metal Corrosion Resistant Bioinspired Special Wetting Surface Based on Laser Texturing Technology: A Review"

_micromachines, 2022, doi:10.3390/mi13091431_

Round 1
Reviewer 1 Report
The authors reviewed researches of lase surface texturing technology on preparation of metal superhydrophobic and corrosion-resistant functional surface. This is a very good topic in the field of laser surface modification. However, this paper has a lot of spelling irregularities and logic problems. In particular, the basic knowledge of laser processing and contact angle characterization occupied too much space, which can be found in the reference book. Review article should focus more on the recent researches in this field, rather than putting common sense as the main content. It is suggested that the article should be outlined and re-written before submission.
1. The title of the article is inappropriate. The main content of the article is to popularize some models and characterization methods. It is inconsistent with the title. In particular, there are many kinds of corrosion, which should be specific, not just the electrochemical corrosion mentioned in the abstract.
2. There are a large number of cases of letters and inconsistent formula fonts in the article. The author should try to avoid these primary errors (For example, Page 1, line 41, By contrast, Due to …….).
3. The citation format of references in the table is not uniform.
4. The main contents listed at the end of the paper are different from the actual description. In other words, the article was not written according to this outline, and the logic was quite confusing.
5. The same variables, such as contact angle, are different in the text part from those in the Figure, and the full text should be unified. It is suggested to add an alphabet at the beginning of the article.
6. The schematic diagram of superhydrophilicity should be supplemented in Figure 2.
7. It is very necessary to supplement the schematic diagram of the advance angle and the retraction angle.
8. The schematic diagram of Young’s model should be supplemented in Part 2.2.1.
9. Inconsistent picture definition is a great obstacle to reading.
10. In line 296, why is green laser not involved ?
11. Part 3.2 does not seem to have any effect on the article, and it is suggested to delete it. Part 4 lists too many characterization methods, and the review article should focus more on previous studies and comments.
Author Response
We have revised the manuscript and replied the comments,please see the attachment.

Reviewer 2 Report
(1) In table 1, coatings by spraying or painting are missing. Obviously, coating is a predominant way for corrosion resistance. Moreover, we do not believe that the hydrothermal method requires high equipment with poor safety. In a typical example 10.1021/am4000134, just hot water is required.
(2) There is not too much new information in the part of "superhydrophobic mechanism", especially in 2.1.
(3) Is the slippery surface superhydrophobic? If not, why such surface was discussed in this paper entitled "Research on metal superhydrophobic corrosion-resistant functional surfaces based on laser texturing technology: A review"?
(4) As declared in the “Conclusion and outlook”, how to improve the corrosion resistance stability of superhydrophobic surfaces is a very developed field. So, authors should pay more more efforts in this part, rather than the classic wetting model in 2.1. What’s the durability of the common samples? How to improve?
(5) Language tips. The manuscript should be read carefully to avoid grammar errors.
Author Response

(The authors gave the same response as above.)

Round 2
Reviewer 1 Report
The article is recommended to be published after appropriate revision.
Author Response
Thanks for the reviewer's comment.
Reviewer 2 Report
As I have pointed out, the title of this submission is "…superhydrophobic corrosion resistant functional surfaces…". Then, why slippery surface was discussed in this paper? What is the definition of superhydrophobic surface? What's the difference and connection between superhydrophobic surface and slippery surface? Is slippery surface a kind of superhydrophobic surface? Obviously, slippery surface is NOT a kind of superhydrophobic surface. So, you have two ways to solve this obvious paradox, viz., (1) change your title to fit the content, such as "the bioinspired special wetting surface", and (2) delete the related discussion in slippery surface.
Author Response
Thanks for the reviewer's comment, we have modified the article according to the reviewer's suggestion. Please see the attachment.

Round 3
Reviewer 2 Report
Accept, please.